# Effects of Age, Diet CP, NDF, EE, and Starch on the Rumen Bacteria Community and Function in Dairy Cattle

**DOI:** 10.3390/microorganisms9081788

**Published:** 2021-08-23

**Authors:** Yangyi Hao, Yue Gong, Shuai Huang, Shoukun Ji, Wei Wang, Yajing Wang, Hongjian Yang, Zhijun Cao, Shengli Li

**Affiliations:** 1State Key Laboratory of Animal Nutrition, Beijing Engineering Technology Research Center of Raw Milk Quality and Safety Control, College of Animal Science and Technology, China Agricultural University, Beijing 100193, China; B20193040338@cau.edu.cn (Y.H.); gongyue@brightdairy.com (Y.G.); huangshuai510@126.com (S.H.); wei.wang@cau.edu.cn (W.W.); wangyajing_cau@163.com (Y.W.); yang_hongjian@cau.edu.cn (H.Y.); caozhijun@cau.edu.cn (Z.C.); 2College of Animal Science and Technology, Hebei Agricultural University, Baoding 071001, China; jishoukun@163.com

**Keywords:** rumen bacteria, enzyme, fermentation, dietary, age

## Abstract

To understand the effects of diet and age on the rumen bacterial community and function, forty-eight dairy cattle at 1.5 (M1.5), 6 (M6), 9 (M9), 18 (M18), 23 (M23), and 27 (M27) months old were selected. Rumen fermentation profile, enzyme activity, and bacteria community in rumen fluid were measured. The acetate to propionate ratio (A/P) at M9, M18, and M23 was higher than other ages, and M6 was the lowest (*p* < 0.05). The total volatile fatty acid (TVFA) at M23 and M27 was higher than at other ages (*p* < 0.05). The urease at M18 was lower than at M1.5, M6, and M9, and the xylanase at M18 was higher than at M1.5, M23, and M27 (*p* < 0.05). Thirty-three bacteria were identified as biomarkers of the different groups based on the linear discriminant analysis (LDA) when the LDA score >4. The variation partitioning approach analysis showed that the age and diet had a 7.98 and 32.49% contribution to the rumen bacteria community variation, respectively. The richness of *Succinivibrionaceae_UCG-002* and *Fibrobacter* were positive correlated with age (r > 0.60, *p* < 0.01) and positively correlated with TVFA and acetate (r > 0.50, *p* < 0.01). The *Lachnospiraceae_AC2044_group*, *Pseudobutyrivibrio*, and *Saccharofermentans* has a positive correlation (r > 0.80, *p* < 0.05) with diet fiber and a negative correlation (r < −0.80, *p* < 0.05) with diet protein and starch, which were also positively correlated with the acetate and A/P (r > 0.50, *p* < 0.01). The genera of *Lachnospiraceae_AC2044_group*, *Pseudobutyrivibrio*, and *Saccharofermentans* could be worked as the target bacteria to modulate the rumen fermentation by diet; meanwhile, the high age correlated bacteria such as *Succinivibrionaceae_UCG-002* and *Fibrobacter* also should be considered when shaping the rumen function.

## 1. Introduction

Ruminant animals can capture nutrients from roughage by the digestion process of ruminal microorganisms to the cell wall components. The rumen is a complex microbial ecosystem containing a great diversity of bacteria, archaea, viruses, protozoa, and fungi [1,2]. Within the rumen microorganisms, bacteria are the most abundant species and are the major contributor in the digestion of plants [2]. The bacteria can convert the feed into volatile fatty acids (VFA), ammonia, and microbial crude protein (MCP), which could further supply nutrients for ruminants [3,4]. The key role during the degradation of a plant is the enzymes, which were encoded and secreted by the microorganisms [5,6]. The digestive enzyme could catalyze and decompose feedstuff into molecules for animals to use, for example, amylase could decompose the starch into glucose and further enhance the ruminants’ starch digestibility [7]. The exogenous protease could alert the amino acid composition and improve the starch digestibility of corn silage [8]. Bacteria, enzymes, and the VFA, MCP, pH, and NH_3_-N, are closely related and jointly assist in completing the rumen digestive function.

The rumen function and ecosystem stability largely depend on the diversity and complexity of microorganisms [9]. Many studies have illustrated that age and diet could affect rumen microbiota, separately. Fonty et al. found that the rumen cellulolytic bacteria of lambs reach a comparable level of the mature rumen at the end of a week after birth [10]. Jami et al. found that the calf was born with some rumen bacteria essential for mature rumen function [11]. From 6 months to 2 years old, the rumen bacteria community was significantly different with the same diet [11]. Bohra et al. showed the rumen bacteria composition varied with the dietary nutritional level [12]. The different roughage sources also altered the rumen microbiome and carbohydrate-active enzyme profile [12], and the change is associated with the feedstuffs’ nutrients [13]. Besides the independent influence of age and diet on rumen microbiota, rumen microorganisms are also affected by the combined effects of diet and age [14]. For example, the ruminal bacterial community is established before the intake of solid feed, and the increased intake of starter could, in turn, shape this community [15]. Anderson et al. also indicated that, with the solid feed intaking, the proteolytic bacteria increased from 1–2% (at the delivery) to 10% (at 12 weeks); meanwhile, the amylolytic bacteria also increased with age [16]. A meta-analysis showed the bacteria might exert independent effects on various aspects of ruminant performance [17]. Bacteria composition, metabolic pathway, and metabolite also differ with the different milking performance of dairy cows [18]. However, there is still a shortage of specific information about the combined effects of age and diet main nutrients (fiber, protein, fat, and energy) on the rumen bacteria composition and function under the natural feed condition, which limited the precise feeding management of ruminants.

Therefore, this study investigates the ruminal bacteria profile, digestive enzyme activity, and VFAs of dairy cattle in six production stages under different age and diet conditions. We hope to illustrate the rumen bacteria composition and function features under specific feeding stages and find the diet or age-related bacteria and its production (enzymes or VFAs). Ultimately, we hope to provide the theoretical basis for precise dairy cattle feeding and management.

## 2. Materials and Methods

### 2.1. Ethics Statements

The experimental procedures used in the present study were approved by the Ethical Committee of the College of Animal Science and Technology, China Agriculture University (Protocol number: 2013-5-LZ).

### 2.2. Animals and Sample Collection

Animals with a good standard of health, half-sibs, and no antibiotic be used a month before the sampling time were selected from a farm in Beijing, China. Finally, the forty-eight Holstein female dairy cattle were divided into six groups—1.5 (M1.5), 6 (M6), 9 (M9), 18 (M18), 23 (M23), and 27 (M27) months—and each group had eight animals. All the cattle had unlimited access to feed, and the feed was given three times a day. The animal feed formula and chemical composition of these diets are shown in Appendix A. In brief, the M1.5, M6, and M27 had a relatively high diet starch and protein content, while the M9, M18, and M23 had a relatively high fiber diet.

Rumen fluid sample was collected by oral intubation before morning feeding. Approximately 50 mL of rumen liquid from each animal was obtained, with the initial ~50 mL discarded to avoid saliva contamination. Each sample was separated into two sterile tubes. One was immediately placed in liquid nitrogen and stored at −80 °C for 16S rRNA sequencing and enzyme activity analysis. Another was filtered through four cheesecloths and then stored at −20 °C for fermentation profile analysis.

### 2.3. Sample Analysis

#### 2.3.1. Fermentation Profile and Enzyme Activity

The rumen pH was immediately determined after sample collection using a pH electrode (Model pH B-4; Shanghai Chemical, Shanghai, China). The NH_3_-N concentration of rumen fluid was measured using the phenol-sodium hypochlorite colorimetry method described by Broderick and Kang [19]. The MCP concentration was detected according to Makkar et al. [20]. Then, 0.2 mL of 25% metaphosphoric acid was added to 1.0 mL rumen fluid samples, to wipe off the albumen precipitation, before the quantification of VFAs, which were measured by gas chromatography (6890 N; Agilent technologies, Avondale, PA, USA) according to Cao et al. [21]. The urease [22], protease [23], amylase [24], lipase [25], xylanase [25], and dehydrogenase [26] were measured using a SpectraMax 190 Microplate Reader (MD., New York, NY, USA) with the commercial kits (Suzhou Grace Biotechnology Co., Ltd., Suzhou city China); specifically, the rumen fluid was centrifuged at 2500× *g* for 10 min at 4 °C temperature, and the supernatant fluid was ultrasonically broken for 3 min and then centrifuged at 12,000× *g* for 5 min. The measured wavelength of the urease, protease, amylase, lipase, xylanase, and dehydrogenase were 578, 680, 540, 405, 540, and 460 nm, respectively.

#### 2.3.2. 16S rRNA Sequencing

The DNA of rumen fluid samples was extracted using FastDNA SPIN for soil kit (MP Biomedicals, Solon, OH, USA) by centrifuging with the column. DNA concentration and purity were monitored on 1% agarose gels. According to the concentration, DNA was diluted to 1 ng/µL using sterile water. The V3–V4 region of the 16s rRNA gene was amplified by PCR (denaturation: 94 °C for 2 min followed by 30 cycles at 98 °C for 10 s, annealing reaction: 62 °C for 30 s; 68 °C for 30 s; and a final extension at 68 °C for 5 min) using specific primer: former primer 341F (CCTACGGGNGGCWGCAG), reverse primer 806R (GGACTACHVGGGTATCTAAT) [27]. Amplicons were extracted from 2% agarose gels and purified using the AxyPrep DNA Gel Extraction Kit (Axygen Biosciences, Union City, CA, USA) according to the manufacturer’s instructions. The amplicons were quantified using an ABI StepOnePlus Real-Time PCR System (Life Technologies, Foster City, CA, USA). The purified amplicons were pooled in equimolar and paired-end sequenced on a PE250 Illumina platform. Paired-end reads were merged using FLASH (V1.2.7, http://ccb.jhu.edu/software/FLASH/, accessed on 1 October 2017) [28]. Low-quality reads, such as reads with length < 200 bp, containing ambiguous bases, or unmatched to primer sequences and barcode tags, were filtered to obtain the high-quality clean tags [29] according to the QIIME (V1.9.1, http://qiime.org/scripts/split_libraries_fastq.html, accessed on 14 May 2018) [30] quality-controlled process. The tags were compared with the reference database (Silva database, https://www.arb-silva.de/, accessed on 1 September 2019) using the UCHIME algorithm (UCHIME Algorithm, http://www.drive5.com/usearch/manual/uchime_algo.html, accessed on 1 October 2017) [31] to detect chimera sequences. Then, the chimera sequences were removed [32], and the effective tags were finally obtained. Sequence analysis was performed using Uparse software (Uparse v7.0.1001, http://drive5.com/uparse/, accessed on 1 October 2017) [33]. Sequences with ≥97% similarity were assigned to the same OTUs. The representative sequence for each OTU was screened for further annotation. OTUs abundance information was normalized using a standard of sequence number corresponding to the sample with the least sequences (OUT number = 26,700). Subsequent analysis of alpha-diversity and beta-diversity was performed basing on this output normalized data. For each representative sequence, the Silva Database 132 (http://www.arb-silva.de/, accessed on 1 September 2019) [34] was used based on a Mothur algorithm to annotate taxonomic information.

### 2.4. Statistics

The rumen fermentation profile and enzyme activities were subjected to one-way ANOVA by SAS (version 9.4, SAS Institute Inc., Cary, NC, USA). Alpha-diversity indices were calculated with QIIME (Version 1.7.0) and analyzed using the Kruskal–Wallis test and Wilcoxon rank test using the “dplyr” package in R. Principal Co-ordinates Analysis (PCoA), and analysis of similarities (ANOSIM) (999 permutations) was performed and visualized using the “ggplot2” package in R (Version 3.6.1). Spearman’s rank correlation was used to identify the relationship between the enzyme activity and rumen fermentation profile (VFA, NH_3_-N, and MCP); the top 50 abundant bacteria at genus level and its byproducts (enzyme, VFA, NH_3_-N, and MCP) were identified using the “corrplot” package in R. The result was visualized as a heatmap using the R package “pheatmap.” All *p*-values were corrected using a false discovery rate of 0.05, as described by Benjamini and Hochberg [35]. The false discovery rate corrected *p* < 0.05 was considered significant. The linear discriminant analysis effect size (LEfSe) [36] was used to determine the difference of rumen bacteria among ages and diets by coupling Kruskal–Wallis Test for statistical significance with additional tests assessing biological consistency and effect relevance. Variation partitioning approach (VPA) was used to evaluate the relative importance of age and dietary nutrients on rumen bacteria community using the “vegan” package in R [37]. Spearman’s rank correlation and liner regression were also used to analyze the relationship between the PC1 (principal component 1 of the principal coordinate analysis’ axis of rumen bacteria) and age or diet nutrients [38]. 

## 3. Results

### 3.1. Rumen Fermentation Profile and Enzyme Activity

The rumen pH among the M1.5, M6, M9, and M18 dairy cattle did not differ, while the ruminal pH value in these groups was higher than in the M27 (*p* < 0.05) (Table 1). The NH_3_-N content in M1.5 was the highest (*p* < 0.05). The MCP content in M6, M23, and M27 was higher than in M1.5 and M9, and it was the lowest (*p* < 0.05) in M18. The acetate concentration in M9, M18, and M27 did not differ, while that in M1.5 was lower than in other groups, and that in M6 was also significantly lower than in M18 (*p* < 0.05). Propionate content at M6 and M27 was higher than that in M1.5 and M18 (*p* < 0.05). The rumen butyrate concentration at M1.5 was lower than in others (*p* < 0.05). The rumen butyrate in M27 and M6 was significantly higher than in M18 (*p* < 0.05). The total VFA concentration in M27 and M23 were higher than in others (*p* < 0.05). The acetate to propionate ratio (A/P) in M23, M18, and M9 was higher than in other ages, and it was lowest (*p* < 0.05) in M6.

The dehydrogenase in the M6, M9, and M27 groups was higher than in M1.5 (*p* < 0.05), while M9, M18, M23, and M27 had no difference (Table 2). The urease in M18 was lower than in M1.5, M6, and M9 (*p* < 0.05). The protease in M1.5 was higher than the M9 (*p* < 0.05). The xylanase in M18 was higher than in M1.5, M23, and M27 (*p* < 0.05). The amylase in M6 and M27 was higher than in M1.5 and M9 (*p* < 0.05). The lipase in M6 was higher than in M1.5, M9, M18, and M23 (*p* < 0.05), while that in M9, M18, M23, and M27 did not differ.

Spearman’s rank correlation was performed to study the correlation between enzyme activity and rumen fermentation profile (VFA, NH_3_-N, and MCP) (Appendix A). As a result, we observed that dehydrogenase and amylase were positively correlated with propionate and valerate (r > 0.50, *p* < 0.01). Xylanase was negatively correlated with NH_3_-N (r = −0.56, *p* < 0.01). Rumen amylase negatively correlated with A/P (r = −0.57, *p* < 0.01).

### 3.2. Rumen Bacteria Analysis

#### 3.2.1. Rumen Bacteria Diversity Analysis

After sequence trimming, quality filtering, and chimeras removing, a total of 2,575,670 high-quality sequence tags was obtained from all samples. The M1.5, M6, M9, M18, M23, and M27 groups had 453,635 (56,704 ± 1418, mean ± standard deviation), 448,576 (56,704 ± 1803), 408,982 (51,122 ± 1468), 418,263 (52,282 ± 1800), 411,811 (51,485 ± 1583), and 434,270 (53,658 ± 1986) tags, respectively (Appendix A). Good’s coverages for all samples were more than 99.70%. The alpha-diversity indices, including Chao1, ACE, observed OTUs, Shannon, and Simpson index, were compared among six groups (Appendix A). Interestingly, the observed OTUs, ACE, and Chao1 values in M18 and M23 were significantly higher than in M6 and M9; those in M1.5 and M6 were lower than in others (*p* < 0.05). The Shannon index was increased from M1.5 to M18 but showed no difference between M18 and M23. The Shannon index of M27 was lower than that of M18 and M23 (*p* < 0.05). The Simpson index of M1.5 was higher than that of other groups (*p* < 0.05). These indexes show that M18 and M23 had the highest bacteria diversity.

The Venn diagram analysis revealed that 1113 operational taxonomic units (OTUs) were shared across the six groups (Figure 1A). There are 261, 186, 182, 292, 269, and 335 unique OTUs in M1.5, M6, M9, M18, M23, and M27, respectively. The PCoA analysis showed that M1.5 and M6 separated with others (Figure 1B), and ANOSIM showed these groups were statistically different (R^2^ = 0.62, *p* = 0.001).

#### 3.2.2. Rumen Bacteria Composition Analysis

The top ten phyla account for more than 99.9% of bacteria (Appendix A). Twenty-two genera were identified as core bacteria, which were identified with a relative abundance >1% and present in at least 80% of all samples (Appendix A). Bacteria with LDA scores higher than four were speculated to have a different abundance across the different groups (Figure 2A). Finally, 33 bacteria were identified as biomarkers of the various groups, respectively. The unique bacteria in M1.5 were Proteobacteria, Gammaproteobacteria, Succinivibrio, Lachnospiraceae, and Bacteroidaceae (genus level). Prevotellaceae, Veillonellaceae, Selenomonadales, Negativicutes, and Muribaculaceae were higher in M6. Prevotella_ruminicola and Lachnospiraceae (family level) were higher in M9. Some Firmicutes phylum bacteria could be the biomarker in M18 (Figure 2B), such as Clostridia, Firmicutes, Ruminococcaceae, Christensenellaceae, Rikenellaceae, and Bacteroidales. The unique bacteria at M23 were Fibrobacter. Succinivibrionaceae and Aeromonadales were higher at M27 (*p* < 0.05).

### 3.3. Driving Factors and the Correlations between Rumen Bacteria and Its Byproducts

#### 3.3.1. Driving Factors of Rumen Bacteria Variation

Variation partitioning approach (VPA) revealed diet and age factors explained 32.49 and 7.98% of rumen bacteria communities’ variations (Figure 3A). The CP, NDF, starch, and EE had a 4.50%, 4.31%, 4.64%, and 5.44% effects on the rumen bacteria community (Figure 3B). The Spearman rank correlation analysis showed that age and NDF negatively correlated with PC1 (r = −0.66 and −0.83, *p* < 0.01, respectively). CP, starch, and EE positively correlated with PC1 (r = 0.83, 0.67, and 0.65, *p* < 0.01, respectively) (Appendix A).

#### 3.3.2. The Correlation between Bacteria and Its Main Byproducts

To explore the potential roles of ruminal bacteria on enzyme activity and fermentation profile, we analyzed the relationship between the top 50 abundant genera and their main byproducts (enzyme, VFA, NH_3_-N, and MCP) using Spearman’s correlation analysis (Figure 4). We found that 23 bacteria were significantly correlated with A/P, acetate, and TVFA (*p* < 0.05). Five genera belong to phyla Firmicutes, and eight genera belong to phyla Bacteroidota. Additionally, 13 bacteria genera were significantly correlated with NH_3_-N, valerate, and urease (*p* < 0.05). Four genera belong to the Firmicutes phyla; five genera belong to the Bacteroidota phyla; two genera belong to the Proteobacteria phyla. The genera of Shuttleworthia, Oribacterium, Prevotellacear_YAB2003_group, and Succinivibrionaceae_UCG-001 were positively correlated (r > 0.5, *p* < 0.05) with the dehydrogenase, isovalerate, MCP, and propionate, while genera of Prevotellaceae_NK3B31_group, UCG-005, Butyrivibrio, and Rikenellaceae_RC9_gut-group were negatively correlated (r < −0.5, *p* < 0.05) with them. Specifically, the genus of Succinivibrionaceae_UCG-002, Treponema, and Eubacterium_ruminantium_group were in strong positive correlation with acetate (r > 0.73, *p* < 0.01).

The Spearman’s correlation coefficient of the top 50 genera with age or diet is in Appendix A. We selected five bacteria genera highly correlated with age and diet and correlated with the VFA (Figure 5). Succinivibrionaceae_UCG-002 and Fibrobacter are positively correlated with age (r > 0.60, *p* < 0.01) and positively correlated with TVFA and acetate (r > 0.50, *p* < 0.01). Lachnospiraceae_AC2044_group, Pseudobutyrivibrio, and Saccharofermentans have a Spearman’s correlation coefficient value >0.80 with diet NDF and <−0.80 with diet CP and starch (*p* < 0.01), which also positively correlates with the acetate and A/P (r > 0.50, *p* < 0.01). These bacteria should be targeted when regulating the rumen function based on different ages and diet backgrounds. 

## 4. Discussion

### 4.1. Rumen Fermentation Profile and Enzyme Activity

Rumen pH was affected by the diet’s chemical composition, and high dietary NDF content could increase the rumen pH (Jiang et al., 2017), while the high grain diet could produce more fatty acids and further reduce the rumen pH [39]. The M1.5 group received the lowest NDF and highest grain content diet (Appendix A). Still, the incomplete rumen function could not produce enough fatty acids making the rumen pH decreased. VFAs are the end products of diets’ fermentation, and they are also essential for rumen development, production performance, and body metabolism [40,41,42]. Previous studies showed that diet chemical composition could alter the rumen VFA production [21,41,42]. High diet starch content could enhance the rumen propionate concentration [41]. The calf at the age of 1.5 months had a lower propionate concentration was due to the immature rumen function, which could not produce enough enzyme to degrade the starch into propionate. In M18, dairy cattle with the lowest diet starch content also had less propionate, caused by the lack of substance, such as starch. A high fiber content diet could enhance the rumen acetate concentration and the A/P value [43]. Additionally, the A/P is age-related [44]. The M27 group had a lower A/P value than M9 and M18, and the discrepancy suggested that the diet takes on a more important role in shape rumen fermentation.

Non-protein nitrogen could be hydrolyzed into ammonia by urease produced by microbes [45]. The protein is hydrolyzed into amino acids and peptides by protease, and then parts of amino acids also became ammonia by microbial deamination [45]. A portion of ammonia synthesize MCP via microorganisms, and the other parts are absorbed into the blood, participating in the rumen nitrogen cycle [46]. Our results indicated that, with the high protein diet, unmatured rumen absorption function [47] in M1.5 lead to the high NH_3_-N content in the rumen. MCP had a significantly important role in the ruminants’ production performance and diet CP utilization efficiency. In our study, low CP and energy levels in diets inhibited rumen synthesis of MCP (M9 and M18) [48]. Our results demonstrated that dietary protein level, enzyme activity, and matured rumen function were three critical factors for rumen utilization of protein.

### 4.2. Rumen Bacteria Composition

Although the rumen bacteria community has been established in the calf period, the change in rumen bacteria is still age-related in 6 to 120 months [44]. The observed OTUs and diversity index were increased with age; however, the decrease between M23 and M27 indicated that the dietary had a more decisive influence on rumen bacteria diversity. The transition of rumen bacteria between M23 and M27 was consistent with Zhigang et al. [49], who indicated the change from a high fiber to a low fiber diet decreased rumen bacteria diversity. Jami et al. founded that the rumen bacteria community was affected by age and diet [11]. The genus with a relative abundance >1% and present in at least 80% of all samples was defined as core bacteria. The core bacteria were established during the calf stage and testified by Appendix A. However, under the specific age and diet condition, rumen cultured unique genera to finish the particular rumen function in this stage.

Our result showed that the *Gammaproteobacteria* was rich in M1.5. This is consistent with Rey et al., who stated that the *Gammaproteobacteria* was the dominant bacteria (24% relative abundance) in calf at the age of 15–83 days [50]. Firmicutes strongly correlated with fiber digestion and could degrade complex carbohydrates, such as cell surface [51,52]. *Firmicutes*, *Clostridia*, and *Ruminococcaceae* were rich in M18, digesting the high fiber diet [52,53]. Huws et al. indicated that *Fibrobacteria* was abundant in the rumen bacteria community under the ryegrass diet, which also plays a vital role in forage degradation [54]. M23 had different fiber-correlated bacteria, such as *Fibrobacteria*, from M18: this is because the roughage type affected these bacteria [55]. Our results also found that the rumen bacteria composition was concerned with the nutrient level and the feedstuff species [55,56]. Diet supplement with nitrate could increase *Succinivibrio*, which worked efficiently in the nitrogen utilization [57,58]. Under the high CP diet condition, M27 was rich in *Succinivibrionaceae*, while M1.5 and M6 were not. This increase in *Succinivibrionaceae* was also age-related [44]; it may reach a certain abundance under the specific age and dietary conditions to come into play.

### 4.3. The Relationship within the Rumen Bacteria, Enzyme, and VFA

Rumen was the most important workshop for the digestion of the nutritional substance of ruminants. Bacteria play a crucial role in digesting and converting plant materials to VFA and MCP [59]. The enzyme, which was secreted by bacteria, could catalyze feedstuff decomposition and nutrient turnover [5]. The acetate, TVFA, A/P, NH_3_-N, urease, valerate, and xylanase strongly correlated with rumen bacteria in our study. Bacteria act as processors to connect the diets and these end products. The genus *Fibrobacter* plays a vital role in cellulolytic and converts feeds into VFA [54,60]. *Pseudobutyrivibrio* could degrade the complex plant polysaccharides and produce VFA for ruminants to utilize [61]. *Saccharofermentans* belong to the *Bacteroidetes* phylum, including 116 genes encoding glycosyl hydrolases involved in hemicellulose, pectin, arabinogalactan, starch, fructan, and chitin degradation [62]. These age- or diet-related genera could work as the target bacteria to regulate the rumen function under different feeding backgrounds. The age-related bacteria affected the TVFA and acetate, while the diet-related bacteria affected the A/P and acetate. From the age and diet-related bacteria and their relationship with TVFA and A/P, it can be concluded that the diet could change the rumen fermentation type. In contrast, age influences the rumen fermentation ability.

## 5. Conclusions

Although the rumen bacteria community has already been established at the calf stage, the rumen bacteria composition still changes along with age and diet variation. This study gave the quantitative effect of diet (CP, NDF, EE, and starch) and age on the rumen bacteria (explained 32.49 vs. 7.98% bacterial community variation, respectively). Comprehensive correlations were observed between rumen bacterial community, microbiota functions, and rumen fermentation capacities. Our results reveal targeting the bacterial community by diet to regulate rumen fermentation is an efficient method, but dairy cattle’s age should also be considered. Besides diet (CP, NDF, EE, and starch) and age, there are more unknown factors that affect the rumen bacteria community of dairy cattle, which need to be further explored.

## Figures and Tables

**Figure 1 microorganisms-09-01788-f001:**
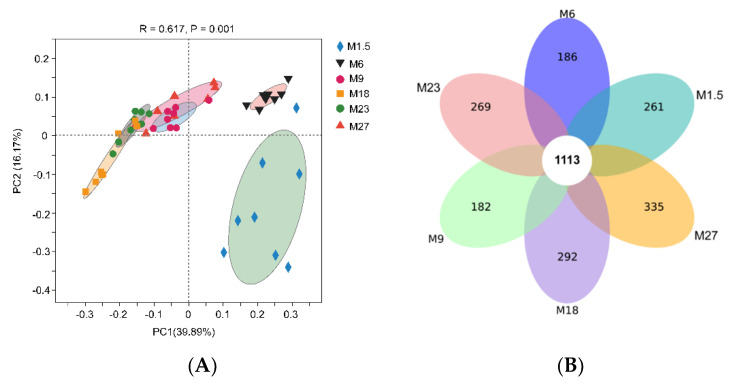
(**A**) Number of shared and unique bacterial OTUs (operational taxonomic units) within each age across all stages of dairy cattle. (**B**) Principal coordinate analysis (PCoA) with Bray–Curtis dissimilarity of the rumen microbial community in different stages of dairy cattle at the genus level. M1.5: age 1.5 months dairy cattle; M6: age 6 months dairy cattle; M9: age 9 months dairy cattle; M18: age 18 months dairy cattle; M23: age 23 months dairy cattle; M27: age 27 months dairy cattle.

**Figure 2 microorganisms-09-01788-f002:**
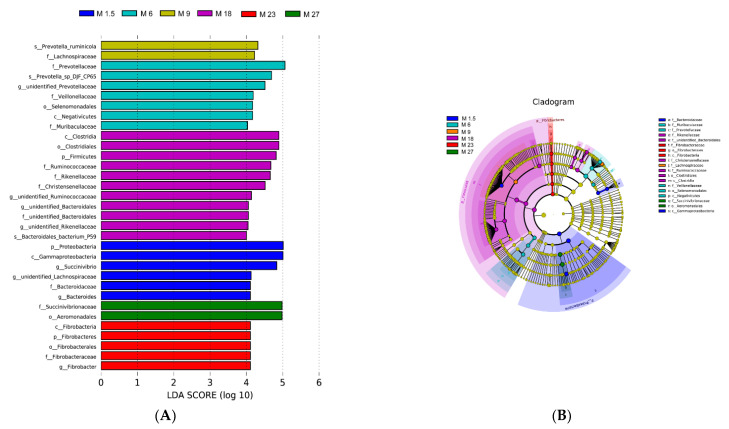
(**A**) Histogram of the LDA scores computed for differentially abundant rumen bacteria across the different ages of dairy cattle. Significant differences are defined as *p* < 0.05 and LDA score >4.0. (**B**) The LDA effect size (LEfSe) analysis of bacterial taxa within the different ages of dairy cattle. Cladogram shows significantly enriched bacterial taxa (from phylum to genus level).

**Figure 3 microorganisms-09-01788-f003:**
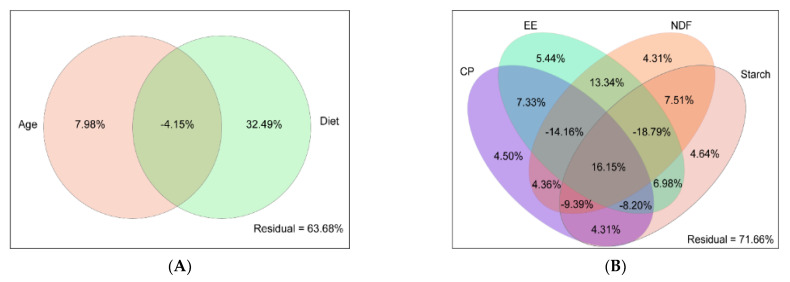
Variation partitioning approach (VPA) analysis the relative importance of (**A**) diet and age, (**B**) CP, NDF, starch, and EE on rumen bacteria composition. CP: crude protein; NDF: neutral detergent fiber; EE: ether extract.

**Figure 4 microorganisms-09-01788-f004:**
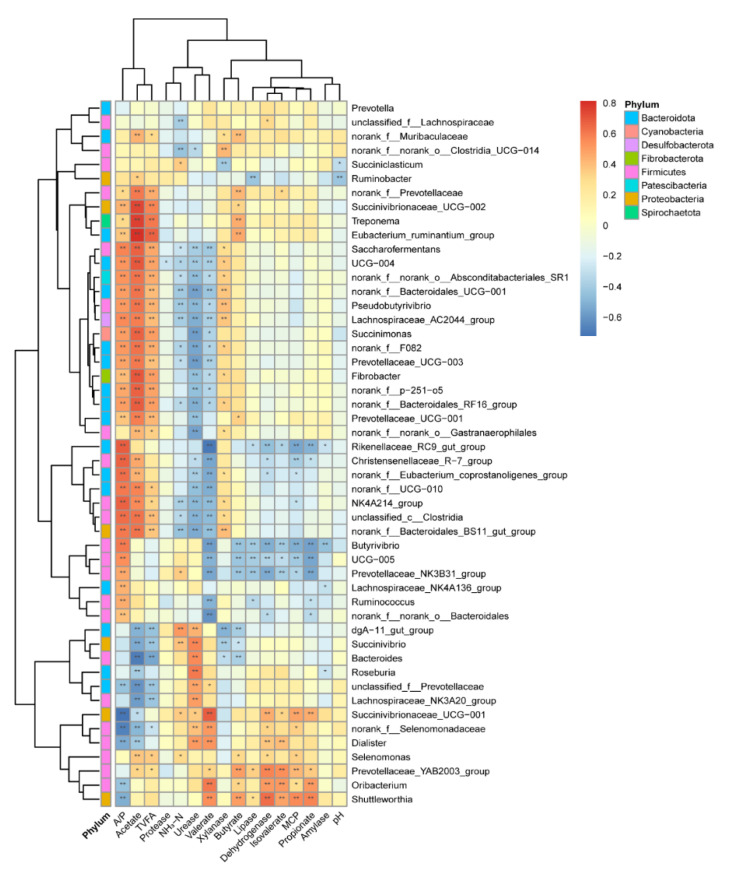
The correlation between bacteria (genus level) and their byproducts. Cells are colored based on Spearman’s correlation coefficient: red represents a positive correlation, and blue represents a negative correlation. “*”, and “**”, indicate FDR (false discovery rate) adjusted *p*-values <0.05 and <0.01, respectively. NH_3_-N: ammonium nitrogen; MCP: microbial crude protein; VFA: volatile fatty acid; TVFA: total volatile fatty acid; A/P: the ratio of acetate to propionate.

**Figure 5 microorganisms-09-01788-f005:**
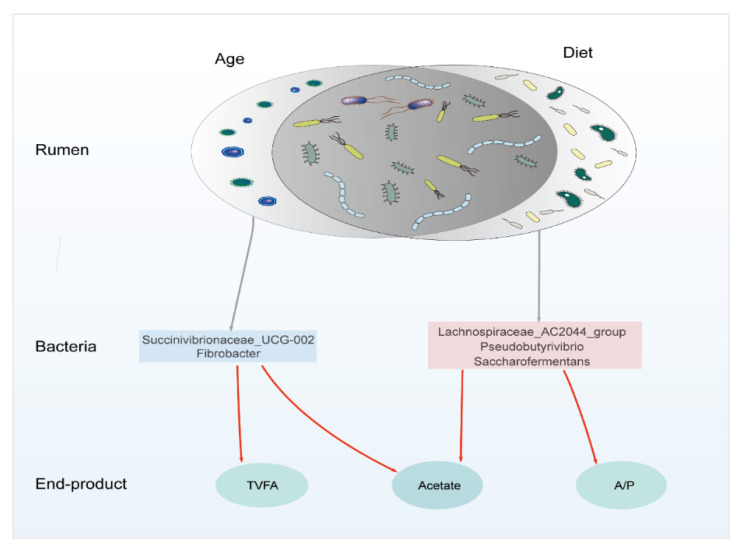
Diagram of the proposed mechanism for age and diet influence on the rumen function. All these bacteria were at the genus level and significantly correlated with TVFA, acetate, and A/P (r > 0.50, *p* < 0.01). Succinivibrionaceae_UCG-002 and Fibrobacter were correlated with age (r > 0.60, *p* < 0.01). Lachnospiraceae_AC2044_group, Pseudobutyrivibrio, and Saccharofermentans have a Spearman’s correlation coefficient value >0.80 with diet NDF and <−0.80 with diet CP and starch (*p* < 0.01). The red line indicates that the bacteria positively correlate with TVFA, acetate, and A/P. Our results give the target bacteria to regulate the rumen function based on different age or diet conditions, promising to provide a theoretical basis for the precision feeding of dairy cattle. TVFA: total volatile fatty acid, CP: crude protein, NDF: neutral detergent fiber, A/P: acetate to propionate ratio.

**Table 1 microorganisms-09-01788-t001:** The rumen fluid fermentation profile of dairy cattle.

Items	Groups						SEM	*p*-Value
	M1.5	M6	M9	M18	M23	M27		
pH	6.52 ^ab^	6.69 ^a^	6.50 ^ab^	6.53 ^ab^	6.36 ^bc^	6.25 ^c^	0.04	<0.01
NH3-N (mg/dL)	32.98 ^a^	11.23 ^de^	13.61 ^cd^	8.97 ^e^	16.41 ^c^	20.35 ^b^	1.22	<0.01
MCP (µg/mL)	81.05 ^b^	116.01 ^a^	79.14 ^b^	61.62 ^c^	119.29 ^a^	127.07 ^a^	3.96	<0.01
Acetate (mmol/mL)	31.91 ^c^	47.34 ^b^	69.16 ^ab^	75.34 ^a^	73.58 ^ab^	64.55 ^ab^	2.45	<0.01
Propionate (mmol/mL)	13.73 ^d^	24.75 ^a^	18.84 ^bcd^	16.21 ^cd^	20.23 ^abc^	23.24 ^ab^	0.89	<0.01
Butyrate (mmol/mL)	2.50 ^d^	7.48 ^ab^	6.10 ^bc^	5.04 ^c^	6.46 ^abc^	7.76 ^a^	0.32	<0.01
TVFA (mmol/mL)	52.63 ^c^	80.59 ^b^	61.48 ^c^	82.69 ^c^	101.47 ^a^	102.33 ^a^	3.31	<0.01
A/P	3.04 ^b^	1.95 ^c^	3.70 ^a^	3.73 ^a^	3.70 ^a^	3.01 ^b^	0.12	<0.01

SEM: standard error of the mean; M1.5: age 1.5 months dairy cattle; M6: age 6 months dairy cattle; M9: age 9 months dairy cattle; M18: age 18 months dairy cattle; M23: age 23 months dairy cattle; M27: age 27 months dairy cattle. NH_3_-N: ammonium nitrogen; MCP: microbial crude protein; TVFA: total volatile fatty acid; A/P: the ratio of acetate to propionate. The different superscript letters mean the difference is significant (*p* < 0.05).

**Table 2 microorganisms-09-01788-t002:** The enzyme activity in rumen fluid of dairy cattle.

Items	Groups						SEM	*p*-Value
	M1.5	M6	M9	M18	M23	M27		
Dehydrogenase (µg/min/mL)	0.49 ^c^	0.87 ^a^	0.79 ^ab^	0.64 ^bc^	0.63 ^bc^	0.77 ^ab^	0.03	<0.01
Urease (µg/min/mL)	2.63 ^a^	2.15 ^abc^	2.31 ^ab^	1.36 ^d^	1.68 ^cd^	1.85 ^bcd^	0.09	<0.01
Protease (µg/min/mL)	14.81 ^a^	10.12 ^ab^	9.34 ^b^	12.27 ^ab^	10.54 ^ab^	13.84 ^ab^	0.69	<0.01
Xylanase (nmol/min/mL)	131.68 ^c^	183.26 ^bc^	250.43 ^ab^	311.81 ^a^	214.75 ^b^	221.89 ^b^	12.20	<0.01
Amylase (mg/min/mL)	0.57 ^cd^	1.10 ^ab^	0.31 ^d^	0.85 ^bc^	0.48 ^bc^	1.47 ^a^	0.17	<0.01
Lipase (nmol/min/mL)	101.49 ^c^	145.07 ^a^	117.48 ^bc^	113.65 ^bc^	114.30 ^bc^	127.25 ^ab^	3.15	<0.01

SEM: standard error of the mean. The different superscript letters mean the difference is significant (*p* < 0.05).

## Data Availability

The datasets analyzed are not publicly available due to ownership by the funding partners, but are available from the corresponding author on reasonable request.

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
