# Peer review of "Effects of Age, Diet CP, NDF, EE, and Starch on the Rumen Bacteria Community and Function in Dairy Cattle"

_microorganisms, 2021, doi:10.3390/microorganisms9081788_

Round 1
Reviewer 1 Report
All comments and suggestions were accepted by the Authors, the suggested changes and corrections were made, and they are appropriate and satisfactory.
However, a few more comments or rather replices are necessary now yet.
Heifers after first calving are primiparous cows, it is a commonly used term.
The Authors replied:
“All the animals were fed on a scale farm and have free access to water and feed ad libitum. All of them were fed three times a day and the light time was the same. Therefore we said they were under the same management system.”
This is somewhat incorrect: were fed ad libitum and fed three times a day? Perhaps better: animals had unlimited access to feed, the feed was given three times a day.
The review says: Is it necessary to provide a description of the groups under each graph? It was only the question of whether these descriptions are needed into each graph.
Author Response
Dear reviewer:
Thanks for your comments to let us better understand the reviewer's meaning and make some careful corrections. Here are some replies to your comments.
The Authors replied:
Point1: “All the animals were fed on a scale farm and have free access to water and feed ad libitum. All of them were fed three times a day and the light time was the same. Therefore we said they were under the same management system.”
This is somewhat incorrect: were fed ad libitum and fed three times a day? Perhaps better: animals had unlimited access to feed, the feed was given three times a day.
Reply:
Dear reviewer: thanks for your kindly correction and we have corrected the sentence as follows “All the cattle had unlimited access to feed, the feed was given three times a day.” And this part also was highlighted in blue color. Thanks for your help to improve our manuscript sentence.
Point2:The review says: Is it necessary to provide a description of the groups under each graph? It was only the question of whether these descriptions are needed into each graph.
Reply:
Dear reviewer: we were sorry to misunderstand your means. We have deleted the unnecessary description of groups and just reserve it under Table 1 and Figure 1.
Thanks again for your work.
Best wishes
Reviewer 2 Report
Authors did not include the majority of my comments in the revised form of their article.
I think that the corrections are necessary in order to substantially improve the article.
Author Response
Dear reviewer:
Thank you again for your comments and we have replied to your comments in detail. Please find it in the attachment.
Best wishes
Yangyi Hao

This manuscript is a resubmission of an earlier submission. The following is a list of the peer review reports and author responses from that submission.
Round 1
Reviewer 1 Report
Recommendation: The above paper is not suitable for publication in its present form.
General comment
The article provides useful information about the effects of diet and age on rumen bacteria community and function in dairy cattle. Although the experiment is in general appropriately designed and implemented, there are some points that should be corrected or clarified.
Major comments
- Please remove number of reference after the names of the authors and delete the number from the end of the sentence, For example, Page 2 – L5: “Fonty et al. [10] had found that the rumen cellulolytic bacteria of lambs reach at a comparable level of that of the mature rumen at the end of the first week after birth.”
- Authors repeatedly use “indicate”. The verbs “demonstrate, show, find, suggest etc” could alternatively be used. At the same time, after these verbs the word “that” should be added. For example, Page 2 – L10: “Anderson et al. [13] also indicated that proteolytic bacteria increased from 1-2% (at the delivery) to 10% (12weeks) as an effect of solid feed ingestion; meanwhile, the amylolytic bacteria were also increased with age.”
- Page 4: Please check the sentence referring to acetate because it is not correspond to the data presented in Figure 1.
Minor points (No Line numbering, so please check in the attached pdf file)
Reviewer 2 Report
In the last decade, changes in the microbiome of the ruminant digestive tract, resulting in changes in the rumen fermentation profile, and consequently, in the level of dairy performance traits, have become the focus of research conducted by many teams. For this reason, it can be stated that the work presented for evaluation brings new elements, and importantly, indicates a need for further research in this subject area.
I cannot fully agree with the statement that the influence of diet and age has not been analyzed so far. An example would be a work published by Kumar et al. (2015) (https://doi.org/10.3389/fmicb.2015.00781).
Generally, there are no methodological objections, but some questions need to be clarified.
Were all groups of animals really kept under the same conditions?
Maybe it is worth adding, what was the sex of the animals, are they heifers? Later in the text, the authors write that these are heifers.
What a dairy breed it was?
Are groups M23 and M27 also heifers, or maybe animals after calving?
Were rumen samples taken at the same time? What was the sampling time period?
Position 20 in the reference list is recorded and also cited in the text: Negi et al. (20) ,but the authors are: Makkar H.P.S., Sharma O.P., Dawra R.K., and Negi S.S.
The listing of subsequent items of literature (from 19 to 26) in the scope of the determinations carried out in section 2.3.1 of the methodology is not convincing, although the readers will find all these items, have all the procedures been carried out without any modification? It is worth supplementing this part of the methodology maybe?
Editorial corrections should be made under the graphs, as the description merges with the text below (Figs. 1, 2, 3, 4, 5, 6, 7). Is it necessary to provide a description of the groups under each graph?
The results are discussed in detail, the illustrative material is entirely appropriate and readable, discussion without significant comments or claims, correctly constructed conclusions.
Conclusion:
After a minor revision, little redrafting and supplementation, the work can be published in the Microorganisms in the Gut Microbiota section, but of course, explanations and answers to the questions and comments contained in the review are necessary.
Reviewer 3 Report
This study investigated the effects of age and diets on the rumen bacteria community. Rumen fluid samples are collected from dairy cattle at different ages, and the rumen fermentation and microbiota were characterized using high-throughput sequencing on 16S rRNA genes.
Major comments:
- The objective and its significance are missing in the introduction. The effect of age and diets on the rumen microbiome has been studied in several previous studies. The author need to point out what kinds of questions remained unclear and what is the difference between the current study and previous studies. The hypothesis is missing in the study too.
- It is undoubtedly that rumen fermentation must be related to diets, time, and rumen microbiota. The sample size and sampling frequency might play very critical roles in determining whether the indicators - NH3, TVFA, pH - can represent (reflect) the real condition of rumen fermentation correctly. In reality, the pH in rumen changes instantly between 6.2 to 7 post-feeding. The variation would be neglected if we average the pH values by age for following analysis. So do the other rumen indicators.
- Figure 7, the proposed mechanism, which should be the most important message of this study, is not interpreted in the content.
Minor comments:
- The author should provide more information (species, age, parity) of the animals used in this study.
- The resolutions of most figures (e.g., Figures 1, 2) need to be improved. In addition, the annotations of the superscripts should be included in the figure legends.